# Temperature Compensation Method for Mechanical Base of 3D-Structured Light Scanners

**DOI:** 10.3390/s20020362

**Published:** 2020-01-08

**Authors:** Marcin Adamczyk, Paweł Liberadzki, Robert Sitnik

**Affiliations:** Institute of Micromechanics and Photonics, Faculty of Mechatronics, Warsaw University of Technology, ul. Św. Andrzeja Boboli 8, 02-525 Warsaw, Poland; p.liberadzki@mchtr.pw.edu.pl (P.L.); r.sitnik@mchtr.pw.edu.pl (R.S.)

**Keywords:** 3D scanner, temperature effect, 3D imaging, structured light, mechanical base of 3D scanner

## Abstract

The effect of temperature on three-dimensional (3D) structured light scanners is a very complex issue that, under some conditions, can lead to significant deterioration of performed measurements. In this paper, we present the results of several studies concerning the effect of temperature on the mechanical base of 3D-structured light scanners. We also propose a software compensation method suitable for implementation in any existing scanner. The most significant advantage of the described method is the fact that it does not require any specialized artifact or any additional equipment, nor access to the thermal chamber. It uses a simulation of mechanical base thermal deformations and a virtual 3D measurement environment that allows for conducting virtual measurements. The results from the verification experiments show that the developed method can extend the range of temperatures in which 3D-structured light scanners can perform valid measurements by more than six-fold.

## 1. Introduction

Three-dimensional (3D) structured light scanners are currently used in many applications in the scientific, engineering, and medical domains [1,2,3,4]. They have been used in inverse engineering [2,5] and can be a source of precise 3D models for 3D printing. In medicine, they have been used for spine curvature estimation [6,7], for gait analysis [8], for prosthetics [9,10], and for four-dimensional (4D) measurements of human bodies in motion [11,12,13]. In industry, they have been used to perform quality control [14,15,16], to support the Industry 4.0 intelligent systems [17,18], in augmented reality to support the documentation process of criminal events [19,20,21,22], and in archaeological sites or in cultural heritage conservation workshops as devices used for digitization for archiving or conducting analysis (e.g., analyzing the influence of conservation processes or for the analysis of ageing processes) [16,23,24,25,26,27,28].

The variety of applications of 3D-structured light scanners is so extended as they can provide precise measurements of 3D geometries, often in a short time [4,11,12,13,16,20,21,29]. A typical 3D-structured light scanner consists of 3 main units: a digital camera, a structured light projector, and a rigid mechanical frame which is a support for camera and projector [2,3,15,29,30] (see Figure 1). In front of the scanner, a measurement volume is located. The particular design can vary, depending on the size of the measuring volume and the required measurement uncertainty, speed of measurements, and number of detectors, as well as other factors including aesthetics and performance aspects. As the 3D scanning technique is a fully matured measurement technique, international standards ISO10360-8 [31] and recommendations VDI/VDE 2634 [32] have been proposed, which describe how the metrological validation process should be performed to calculate the maximum permissible error of a length measurement $E_{MPE}$ [15,20,33,34,35].

Many factors affect the accuracy of a 3D-structured light scanner. Some of these are listed in the Ishikawa diagram presented in Figure 2. In general, it is rather easy to indicate the potential sources of errors, but estimating the quantitative contribution of each factor to the overall measurement uncertainty and the final uncertainty of a particular 3D scanner is complicated [15,20,34,35,36].

Among the factors presented in the Ishikawa diagram (Figure 2), the effects of some on scanner uncertainty have not yet been entirely investigated; in particular, we are interested in temperature [15,20,34,37]. The effect of temperature on 3D-structured light scanners can be observed in three aspects: the effect on the digital camera, on the projector, and on the mechanical frame (often called the mechanical base) [2,3,30,38].

The influence of temperature on images captured by digital cameras is an issue which has been described in the literature, and there exist methods for compensation of this effect [39,40,41,42,43,44,45,46].

However, the effects of temperature on the digital camera is not the only factor that needs to be taken into account when considering the effects of temperature on 3D-structured light scanners. The effects of temperature on the projector and the mechanical base of the scanner also need to be investigated. To our knowledge, there are no papers in the literature that have described the effect of temperature on the deformation of images displayed by the projectors used in 3D-structured light scanners. Some papers have described the effect of temperature on the aging effect of a projection matrix [47,48,49] or the effect of temperature on the quality parameters of the displayed images (e.g., contrast, intensity, reproduction of color, color temperature, and signal to noise ratio) [50,51].

Furthermore, there has been no documented scientific research describing the effects of temperature on the mechanical base of 3D-structured light scanners. This issue also requires investigation.

Another approach to the issue of compensation of temperature on 3D-structured light scanners has been described in [52,53]. A distinguishing feature of this method is that it does not investigate the separate effects of temperature on the camera, projector, and mechanical base, but treats the scanner as a whole. The authors of Reference [52,53] used a two-camera photogrammetry system (which is not a 3D-structured light scanner; however, the working principles are similar) and a thermally stable artifact made of invar rods. The authors developed a compensation method and investigated the compensation results when changing the ambient temperature in the laboratory using an air conditioner. We have developed a similar method, but used a thermal chamber to validate our compensation method (as described in Reference [37]). The common denominator of the aforementioned methods is that they require specialized, thermally stable artifacts validated on other more accurate measuring methods. Another issue is the fact that, to perform validation of an advanced compensation method over a wide range of ambient temperatures, access to the thermal chamber is required. These two factors cause these methods to be expensive and often too complex to implement in practice.

There is a group of 3D-structured light scanners in which the influence of temperature on the mechanical base is much larger than that on the camera and projector: Those with a long mechanical base and large measurement volume [6,7,8,12,29]. The measurement volume of this type of scanner can be described as a cuboid, often with a size in the range of 0.5×0.5×0.5–1×2×1 m^3^ (see Figure 3). The spatial resolution is in the range of 0.1–2 mm. The mechanical base of this type of scanner is often designed as a beam or frame, with the camera and projector fixed to its ends [8,12]. Due to the presence of a long mechanical structure, thermal deformations of the mechanical base have a larger influence on scanner uncertainty than the effects of temperature on the camera and projector.

In this paper, we present the outcome of a study concerning the effects of temperature on the mechanical base of 3D-structured light scanners. Furthermore, we propose a software compensation method suitable for implementation in any existing scanner. The significant advantage of the proposed method is the fact that it does not require any specialized artifact or any additional equipment, nor access to the thermal chamber. It uses a simulation of mechanical base thermal deformations and a virtual 3D measurement environment that allows for conducting virtual measurements. Our method is dedicated for scanners with long mechanical bases, but can also be applied to any 3D-structured light scanner and reduce the effects of temperature on its measurement uncertainty.The results from the experiments show that the developed method can extend the range of temperatures in which 3D-structured light scanners can perform valid measurements by more than six times. This study is the continuation of the research introduced in Refs. [37,46], where we described the effects of temperature on the digital cameras used in 3D-structured light scanners and on the calibration accuracy of a full 3D-structured light scanner, as well as proposing a method for their compensation.

## 2. Basics of the Compensation Model

Two of the biggest disadvantages of the compensation methods described in Refs. [37,52] are the necessity of manufacturing a specialized thermally stable artifact and access to the thermal chamber. In this paper, we propose a novel approach to this issue which relies on the simulation of thermal deformations and virtual measurements of a virtual artifact. In this case, the compensation model is developed using synthetic data, but is also valid for a real 3D-structured light scanner. The basics of our methods are shown in Figure 4.

The most critical assumption of our method is that scanner is calibrated at a constant reference temperature after reaching thermal equilibrium (the warming-up process can take up to 1 h, depending on the camera unit and projector unit [39,40,41,42,43,45,46]). The proper calibration process allows calculation of the camera and projector location and orientation in the calibrated co-ordinate system [15,29,55,56,57,58,59] (Figure 4, box 1 to 2a). This position and orientation changes when the mechanical base of the scanner is exposed to varying ambient temperatures. We conduct a set of thermal deformation simulations of the mechanical base using the Finite Element Method (FEM) to calculate the changes in mutual location and orientation between projector and camera (Figure 4, box 2b). Then, this data is used to deform the base of a virtual 3D scanner, which is represents a real 3D scanner (Figure 4, box 3). Using a virtually defined artifact, we are able to calculate the compensation model (Figure 4, box 4). The most significant advantage of our approach is the fact that we do not need any additional specialized artifacts nor a thermal chamber. The compensation model is calculated using data that comes from the simulations. The biggest drawback is the apparent fact that our method does not compensate for the effects of temperature on the camera and projector units. Another inconvenience is the necessity of conducting a precise simulation of the mechanical base thermal deformations.

Our compensation method can be described using the following steps:The real 3D-structured light scanner is calibrated in a constant reference temperature after reaching thermal equilibrium. In our case, the calibration process is the same as that described in Reference [15,29,30,60]. We use a calibration artifact: a flat, white plate with a matte finish with a matrix of round black markers. It is crucial that the CAD model of a 3D scanner unit is required to calculate the compensation model.Using the data from the scanner calibration, we calculate the exact location and orientation of camera unit $\vec{C}\left(T_{C},R_{C}\right)$ and projector $\vec{P}\left(T_{P},R_{P}\right)$ in the calibrated co-ordinate system.The next step is to determine the effect of temperature on the mechanical base of the scanner. To do so, we simulate the thermal deformation of the mechanical base. We use a CAD model of the scanner, as defined in Dassault Catia V5 [61] and use the Generative Structure Analysis workbench. To calculate the effect of thermal deformations of the mechanical base on the location and orientation of projector and camera, we use the VirtualRigidPart functionality of the Generative Structure Analysis workbench [61]. We simulate the thermal deformations of the mechanical base in several ambient temperatures.Knowing the exact location and orientation of camera $\vec{C}$ and projector $\vec{P}$, we create a virtual scanner unit in the 3DsMAX environment [62]. This scanner has the same functionality as the real one. Using the images rendered by $\vec{C}$ and recalculating them in the same software used to control the 3D-structured light scanner, 3DMADMAC [60], we obtain a point cloud of virtually defined objects. This virtual scanner is a true copy of the real scanner.Knowing the changed locations and orientations of camera $\vec{C}\left(\Delta T_{C},\Delta R_{C}\right)$ and projector $\vec{P}\left(\Delta T_{P},\Delta R_{P}\right)$ in a set of different ambient temperatures, we change the locations and orientation of camera and projector of the virtual scanner in 3DsMAX. Then, we put the virtual artifact in the measurement volume of this virtual scanner and perform a set of 3D scans in varying simulated temperatures. As a result, we obtain a set of measurements that represent the virtual artifact in different temperatures of the scanner.Data obtained from measurements of the virtual artifact are used to calculate the temperature compensation model for the real scanner.

Besides the mentioned advantages (the lack of necessity of preparing a specialized artifact or access to the thermal chamber), several other features make the proposed method valuable and easy to implement: It does not require any changes in scanner design, but the 3D scanner needs to be equipped with a temperature sensor. There is no need to modify the control software to use the assessed temperature compensation model. The computed deviation can be easily implemented at the last stage of calculation, when we calculate the co-ordinates of each point.

## 3. The Real 3D-Structured Light Scanner and Its Calibration

To check the correctness of the proposed temperature compensation method for a 3D-structured light scanner mechanical base, we built a prototype (Figure 5), composed of:

-A mechanical base, made of aluminium profile (linear expansion coefficient $\alpha =∼23\times 10^{-6}\left[K^{-1}\right]$) from Bosch Rexroth profile system, with a section of 45 × 45 L and length of 1 m [63];-An FLIR (previous PointGrey) Grasshopper 2.0 GS2-GE-50S5M-C [64] camera with 5 Mpix sensor size and a Fujinon HF25 lens [65];-a digital projector Optoma ML750 [66]; and-two ball heads to adjust the proper orientation of camera and projector.

To calibrate the scanner measurement volume, we used a calibration artifact made of a glass plate with a printed pattern Figure 6. The calibration pattern consisted of a matrix of 7×5 round black markers on a white background. The artifact was validated using the co-ordinate Measuring Machine (CMM) ACCURA 7 by CarlZeiss with the optical scanning probe ViScan [67].

The calibration artifact was placed on a precise linear-rotation manipulator: a linear table (Standa 8MT160-300) [68] and a rotary Table (8MR151) [69]. We processed the calibration procedure using the 3DMADMAC software [29,30,60]. It was divided into two stages: First, the geometrical calibration of the camera is calculated. In the second stage, the absolute phase distribution is calculated as a function of the (X,Y,Z) co-ordinates. It was necessary to capture a set of calibration images (Figure 7) to calculate the complete scanner calibration:

- For geometric camera calibration, a set of 6 images of the artifact placed in front of the camera in positions C I–C VI (Figure 7) were captured. While capturing these images, the projector is just a source of homogeneous ambient lighting.- For phase calibration, a set of images which represent the absolute phase distribution in at least four different calibration artifact positions (positions Ph I–Ph IV) were captured. Each set of phase images also included a set of images with Gray codes for phase unwrapping.

The scanner was calibrated in a laboratory, where the temperature of 24 °C was stabilized by an air conditioner approximately 45 min after initialization. The calibrated measuring volume can be described as a cuboid with dimensions 400×300×100 mm^3^ and spatial resolution of 0.15 mm. To evaluate its uncertainty and estimate the quality of calibration, we used a ball-bar artifact as recommended by VDI/VDE 2634 [32] (Figure 8a). It was built of two ball bearings with a nominal diameter of 40 mm fixed to a marble slab. The ball-bar artifact was validated using a CMM ACCURA 7 by CarlZeiss with the active probe VasGOLD (maximum permissible error for distance measurements for the used CMM was equal to $E_{MPE}=1.7+L/333$
μm; maximum permissible scanning error for the used active probe was equal to $MPE_{Tij}=2.7$
μm).

Table 1 contains the results of ball-bar measurements made by 3D scanner in four different positions (A, B, C, and D; Figure 8b). For each position of the ball-bar artifact, three dimensions were calculated: the distance between the balls *d* [mm], the diameter of the first ball $D_{1}$ [mm], and the diameter of the second ball $D_{2}$ [mm]. By comparing the reference dimensions measured using the CMM with those measured by the 3D scanner, the maximum permissible error was estimated as $E_{MPE}=\pm 0.1$ mm.

By the end of this stage, we had obtained the following data:-The calibrated measuring volume of the 3D-structured light scanner at the reference temperature 24 °C;-the position and orientation of the camera $\vec{C}\left(T_{C},R_{C}\right)$ in the calibrated co-ordination system at the reference temperature 24 °C;-the position and orientation of the projector $\vec{P}\left(T_{C},R_{C}\right)$ in the calibrated coordinated system at the reference temperature 24 °C; and-the evaluated uncertainty of the 3D-structured light scanner, the maximum permissible error $E_{MPE}=\pm 0.1$ mm.

## 4. Simulation of the Effect of Temperature on the Mechanical Base of a 3D Scanner

In this step, we determine the effect of temperature on the mechanical base of the scanner. To do so, we used a Finite Element Method (FEM) and simulated the thermal deformation of a mechanical base of the scanner in Dassault Catia V5 [61] and Generative Structure Analysis workbench. To calculate the effect of thermal deformations of the mechanical base on the location and orientation of the projector and camera, we used the VirtualRigidPart functionality of the Generative Structure Analysis workbench [61] (Figure 9). We simulated the thermal deformations of the mechanical base in several ambient temperatures. In the simulation, we assumed a homogeneous distribution of temperature through the mechanical base. We also assumed that the ball heads for mounting the projector and detector do not deform under the influence of temperature changes. This assumption is an obvious simplification. Nevertheless, the impact of thermal deformations of mounting heads on the change of orientation of the camera and projector is negligibly small, relative to the impact of the deformation of the mechanical base.

Generative Structure Analysis allows for the definition of VirtualRigidParts, which work as perfectly rigid dummy virtual parts and which, in our case, were used to determine the two vectors that represent the orientation and location of the detector and projector, concerning the mechanical base. We defined two RigidVirtalParts:-RigidVirtualPart 1 (camera): a vector and a point that represent the orientation and position of the camera with respect to the mechanical base; and-RigidVirtualPart 2 (projector): a vector and a point that represent the orientation and position of the projector with respect to the mechanical base.

To estimate the effect of varying temperature on the orientation and position of the camera and projector, we ran five simulations, raising the ambient temperature by 5 °C for each subsequent simulation. The obtained results are presented in Table 2. The results show that the mutual orientation of the camera and projector unit remained practically unchanged, while the mutual position indicates a growing distance between them, which is an expected consequence of thermal extension of the mechanical base.

## 5. Virtual Model of a 3D-Structured Light Scanner

The next step was to create a virtual scanner that can be used to determine the impact of the simulated mechanical base thermal deformations of the deformations of a cloud of points obtained with this scanner. For this purpose, we created a virtual scene in Autodesk 3DsMAX [62] that allowed for advanced rendering with a physical camera. The designed scene was composed of a physical camera (the camera unit) and a physical light source, mapped with a bitmap sequence to project a structured light sequence onto the measurement scene. In the measurement volume, there was also a virtual calibration artifact (Figure 10). The position and orientation and every dimension of the virtual scanner and calibration artifact were approximately the same as the real 3D scanner (Figure 10).

The virtual scanner allows for the rendering of a set of bitmaps which can be recalculated with the same software used to control the real 3D scanner [29,60]. In Figure 11, a rendered frame which represents a calibration artifact in one of the calibration locations (Figure 11b) and the real frame of the same artifact position (Figure 11a) as captured by a camera in the scanner are presented. The presented pictures are almost identical; the differences in the co-ordinates describing the centers of the markers are no higher than 0.5 pixels.

The most significant advantage of the designed virtual scanner is the possibility of using various validation artifacts. These artifacts are also virtual, such that they can be of any complexity. In our case, we defined a set of 140 balls with a diameter of 20 mm, evenly distributed in the scanner measuring volume, in layers layers with 35 balls for each layer (see Figure 12). By rendering a set of measurement images, we were able to calculate the point cloud which represents the 140-ball artifact. Then, the calculated cloud of points was segmented into 140 separated clouds; each representing a single ball. After fitting a virtual sphere to the segmented clouds, we were able to calculate the co-ordinates of each ball and compare them to the reference co-ordinates from the virtual scene; this reflects the recommendation described in VDI/VDE 2634 [32] and from which the $E_{MPE}$ error can be calculated. In our case, we wanted not only to calculate the error, but also estimate the effect of mechanical base thermal deformations on the deformation of the scanner’s measurement volume. Using the outcome from thermal simulations and by changing the position and orientation of the camera and projector, we were able to determine this effect.

## 6. The Thermal Correction Model

We calculated a thermal correction model based on data from the simulations and virtual measurements of the validation artifact. The dependent variables utilized in the compensation model included the set of centers of the virtual balls at the reference temperature (before thermal deformation of the mechanical base). The independent variables were the centers of the validation balls for various temperatures of the mechanical base of the scanner. We calculated three *n*-th degree polynomials $Q^{n}$, which described the deformations of the scanner measurement volume caused by the varying mechanical base temperatures. We used a polynomial fit to determine the coefficients associated with the model [70]. We defined three separate polynomials for each of the co-ordinates *X*, *Y*, and *Z*:
$$X_{compensated}=Q_{X}\left(X_{uncompensated},Y_{uncompensated},Z_{uncompensated},T\right),$$
$$Y_{compensated}=Q_{Y}\left(X_{uncompensated},Y_{uncompensated},Z_{uncompensated},T\right),$$
$$Z_{compensated}=Q_{Z}\left(X_{uncompensated},Y_{uncompensated},Z_{uncompensated},T\right).$$

Using the determined thermal correction model, we recalculated the virtual measurements of the validation artifact once more and calculated the compensated centers of each validation balls. To check the quality of the built compensation model, we calculated the distance from the center of the first ball to the other 139 balls and repeated it for subsequent balls, receiving 9660 distances as a consequence. We compared the results to the corresponding distances calculated for the reference temperature before and after compensation. Then, we repeated the calculations for each of the remaining five simulated temperatures.

Figure 13 presents the outcome of these calculations—the relative error of all possible distances between each set of 140 virtual balls in six simulated temperatures, in the form of histograms of distance deviations from the reference temperature.

For reference temperature, all deviations were no higher than ±0.1 mm. This value is the same as the $E_{MPE}$ error for a real scanner, as described in Section 3. When the temperature of the mechanical base changed, the value of the deviation became larger and went out of the range of ±0.1 mm. For the maximum simulated temperature of the mechanical base, ΔT=+25 °C, the value of deviations were up to ±0.6 mm. This issue is directly connected with the deformation of the scanner’s measurement volume, which is caused by the thermal deformation of its mechanical base.

After compensation, the maximum value of relative error (all possible distances between each set of 140 virtual balls as a function of temperature in the form of deviation from the corresponding distances in reference temperature) was no higher than ±0.1 mm in every simulated temperatures.

Figure 14 shows the directions of the measurement scanner’s volume deformations and the quality of the compensation model. The yellow vectors depict the direction of displacement for every second (for better chart readability) virtual ball’s center before compensation caused by the temperature rise of +25 °C; the length of these vectors were multiplied by 20. The blue vectors represent the same directions after applying the compensation model (the length of each blue vector is multiplied by 100).

## 7. Experiments

The calculated model is based on virtual data that comes from simulations and virtual measurements. To validate its usefulness in the case of the real scanner, we conducted several verification experiments. We attached a set of heating resistors to the mechanical base of the real scanner described in Section 3. By passing a current through the set of resistors heat was emitted, which heated the mechanical base of the scanner and caused its deformation. To verify the temperature of the mechanical base, we used the temperature registration unit MultiCon CMC-99 by Simex Ltd [71] with a set of six Pt100 sensors, 4-wired with wire resistance compensation. The temperature sensors were fixed to the mechanical base and covered by black tape to reduce the effect of ambient temperature for temperature measurement. We conducted experiments in a laboratory equipped with an air conditioning system to maintain a stable ambient temperature over the whole experiment. Figure 15 presents the test stand.

We conducted the verification experiments in the following formula:-Before starting, we turned on the power and waited approximately 40 min to ensure that the scanner had reached thermal equilibrium;-we calibrated the scanner, according to the procedure described in Section 3;-we performed the validation procedure recommended by VDI/VDE 2634 [32] with a ball-bar artifact and estimated the $E_{MPE}$ (see Figure 8);-using the calibration data, we extracted the exact calibrated position and orientation of camera and projector unit. Then, we redefined the CAD model for simulations and virtual scanner for virtual measurements (see Section 4 and Section 5);-we simulated the effects for varying temperatures of the mechanical base of the scanner on the deformation of the virtual scanner (as described in Section 4 and Section 5) and calculated the compensation model (described in Section 6);-then, we heated the mechanical base of the scanner with the set of heating resistors and raised the temperature of the mechanical base by approximately $4{\;}^{\circ }$C. After stabilization of the scanner’s mechanical base temperature, we scanned a ball-bar placed in the scanner’s measurement volume;-then, the mechanical base temperature was gradually increased while scanning the ball-bar artifact. As a result, we performed six measurements, increasing the mechanical base to a temperature of approximately $42.8{\;}^{\circ }$C.

## 8. Results

We performed six measurements of ball-bar artifact for six different temperatures of the scanner’s mechanical base. For each temperature, we calculated the distances between the ball-bar balls by fitting a virtual sphere to each segmented point cloud and calculating the co-ordinates of its center. In Table 3 and in Figure 16, the results of the distance measurements before and after applying the compensation model are presented.

Comparing the results before and after compensation, we observe a noticeable improvement. For the highest temperature in the experiment (42.8°, an increment of 18.9° above the reference temperature), the maximum deviation was 0.1788 mm. After applying the compensation model, this value was reduced by 54%. After compensation, all values of deviation for different temperatures were no higher than the $E_{MPE}$ error (±0.1 mm) calculated for the reference temperature.

## 9. Discussion and Conclusions

We have experimentally confirmed the correctness of the presented compensation method. After using the proposed compensation model, the deviations from the nominal distance between the validation balls significantly decreased and were no higher than the deviations determined for the reference temperature. The conducted experiments have confirmed that, in this way, we can reduce the effect of temperature on the mechanical base of the 3D scanner. However, several features may limit the application of the proposed compensation model:The presented compensation method refers only to the effect of temperature on the mechanical base of the 3D scanner. During modeling and simulation, we have treated the camera and projector unit as perfectly rigid elements which are not affected by varying temperatures—an obvious simplification. Our method was initially devoted for use in 3D scanners for which the effect of temperature on the mechanical base is dominant. For this kind of scanner, with a long base distance, our method will present the best results. However, it can also be used for any other 3D-structured light scanner. Taking into account the uncertainty budget of a 3D scanner, it is always an improvement to reduce any source of potential error.We performed the simulation of the temperature effect on the mechanical base with the assumption of uniform temperature distribution over the entire length of the mechanical base. If the temperature distribution over the mechanical base is not uniform (i.e., if there is a temperature gradient), this fact should be taken into account in the simulations. At the same time, we want to emphasize that our simulation applies only to steady-states. It is also essential that the simulation of the temperature effect on the scanner’s mechanical base is only part of the entire compensation method, and can be freely extended. From the proposed compensation method, the only relevant fact is to determine the position of the projector and detector as a function of temperature.To determine the correct model of temperature effect on the 3D scanner mechanical base, we need a faithful model for simulation. Furthermore, correct simulation of the temperature effect on the scanner’s mechanical base should be carried out. In our experience, we have achieved the best results for simple scanner design geometries, in which the mechanical base is in the shape of a supported beam made of a material with a constant coefficient of thermal expansion α and with an isotropic nature of thermal deformation.During the simulation and verification of the model, we only conducted tests with heating of the profile. We were not able to validate operation of the compensation model for a cooled scanner base. The reason for this was a lack of access to appropriate equipment. Nevertheless, given the relatively simple shape of the scanner base, we can risk the statement that, when cooling the base, we can expect symmetric effects to those obtained for heating the base.

The most significant disadvantage of the proposed compensation method, which may significantly reduce the number of potential applications, is the fact that it does not compensate for the effect of temperature on the camera and projector. At the same time, there is a group of 3D scanners for which this method can bring measurable results. An additional advantage of our method is the lack of hardware interference in the scanner design and not needing advanced equipment to build the compensation model. The 3D scanner only needs to be equipped with a temperature sensor which provides information about the temperature of the mechanical base. Access to a thermal chamber and specific validation standards are not required; the calibration standard supplied to the scanner is sufficient.

## Figures and Tables

**Figure 1 sensors-20-00362-f001:**
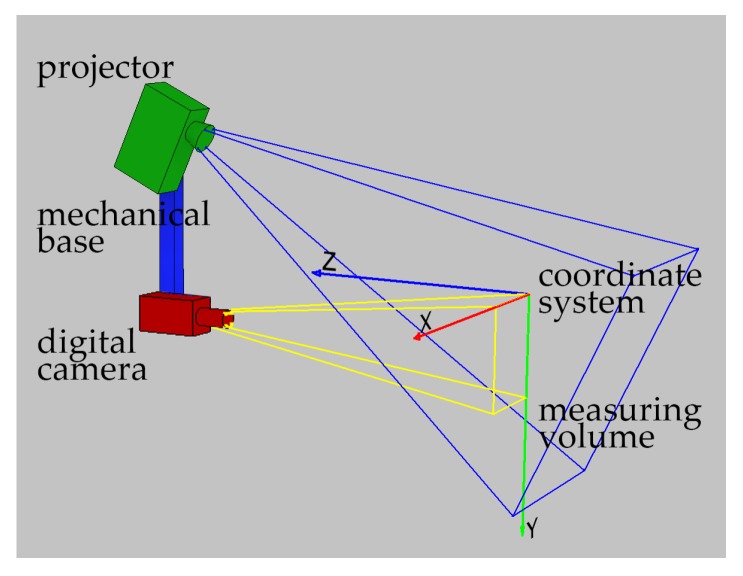
The schematic design of a common 3D-structured light scanner consists of a projector, a mechanical base, and a digital camera.SLSschematic

**Figure 2 sensors-20-00362-f002:**
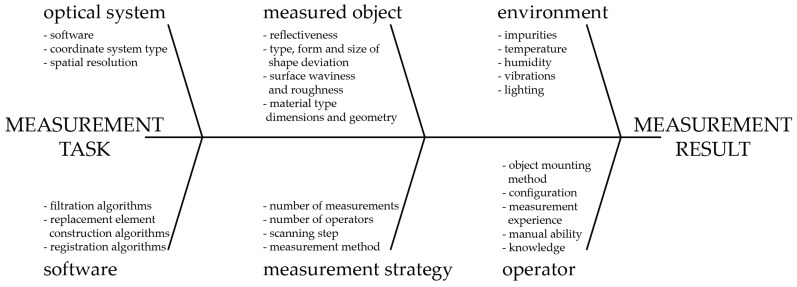
Ishikawa Diagram of the potential external factors which influence the accuracy of a 3D-structured light scanner, based on [34].IshikawaDiagram

**Figure 3 sensors-20-00362-f003:**
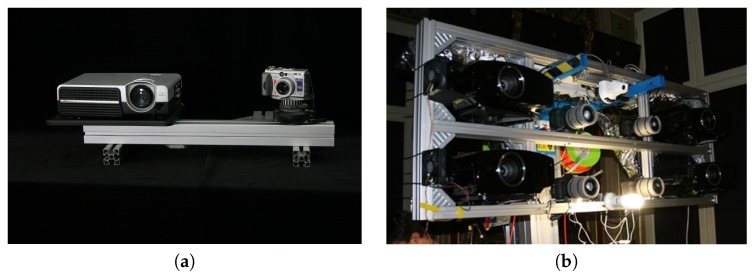
Various design of 3D scanners (**a**) 3D-structured light scanner used in research published in [29]. (**b**) Scanner used for the process of cultural heritage digitization in The Wilanow Palace Museum [54].VariousScanners

**Figure 4 sensors-20-00362-f004:**
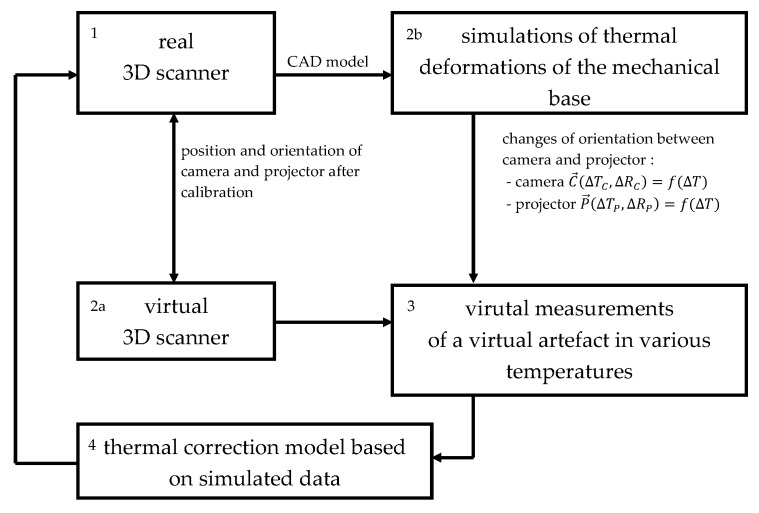
The basics of the proposed temperature compensation method for the mechanical base of 3D-structured light scanners.

**Figure 5 sensors-20-00362-f005:**
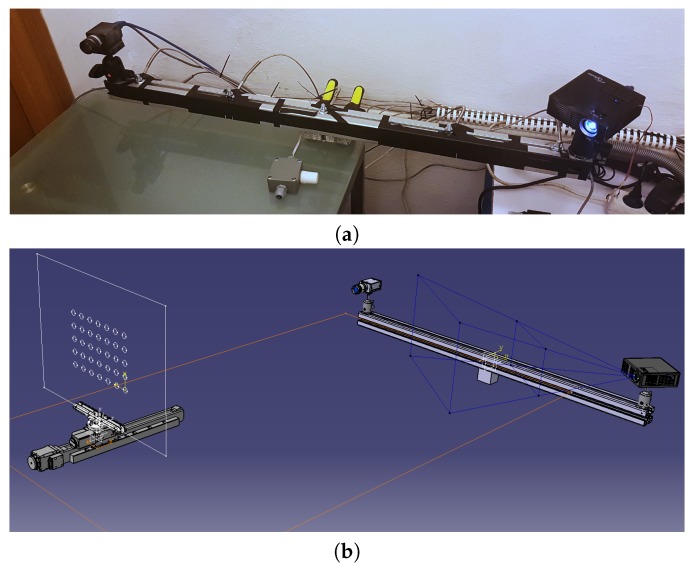
The 3D-structured light scanner designed for verification. (**a**) The mechanical base of this scanner is a beam made from aluminium profile, with the camera and projector fixed to its ends. (**b**) The CAD model of the test stand, with a 3D scanner and calibration artifact.

**Figure 6 sensors-20-00362-f006:**
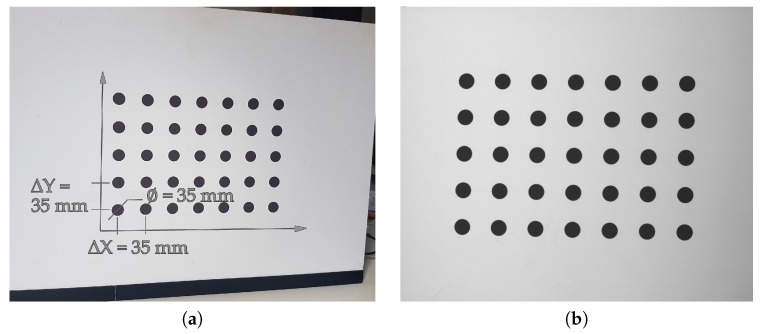
Calibration artifact: (**a**) The calibration artifact is composed of a glass plate with a printed matrix of a 7×5 round markers; and (**b**) a photo of the calibration artifact, as captured by the scanner’s camera.

**Figure 7 sensors-20-00362-f007:**
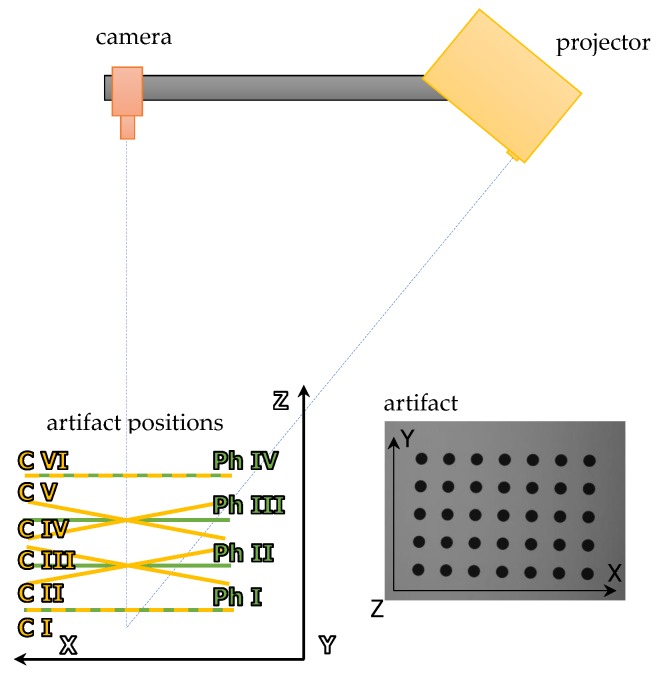
The positions of the calibration artifact required in the calibration process for a 3D-structured light scanner.

**Figure 8 sensors-20-00362-f008:**
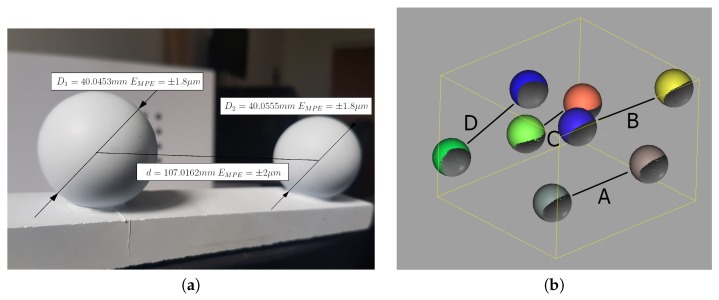
The ball-bar artifact used for scanner validation. (**a**) The ball-bar artifact was previously validated on a co-ordinate Measuring Machine (CMM) to obtain its exact geometry: the exact value of distances between each ball and the diameter of each ball. (**b**) The positions (A, B, C, and D) of the ball-bar artifact in the scanner measuring volume for evaluation of the scanner’s $E_{MPE}$ error. The figure shows four point clouds, which represent the artifact positions and virtual spheres fitted to points.

**Figure 9 sensors-20-00362-f009:**
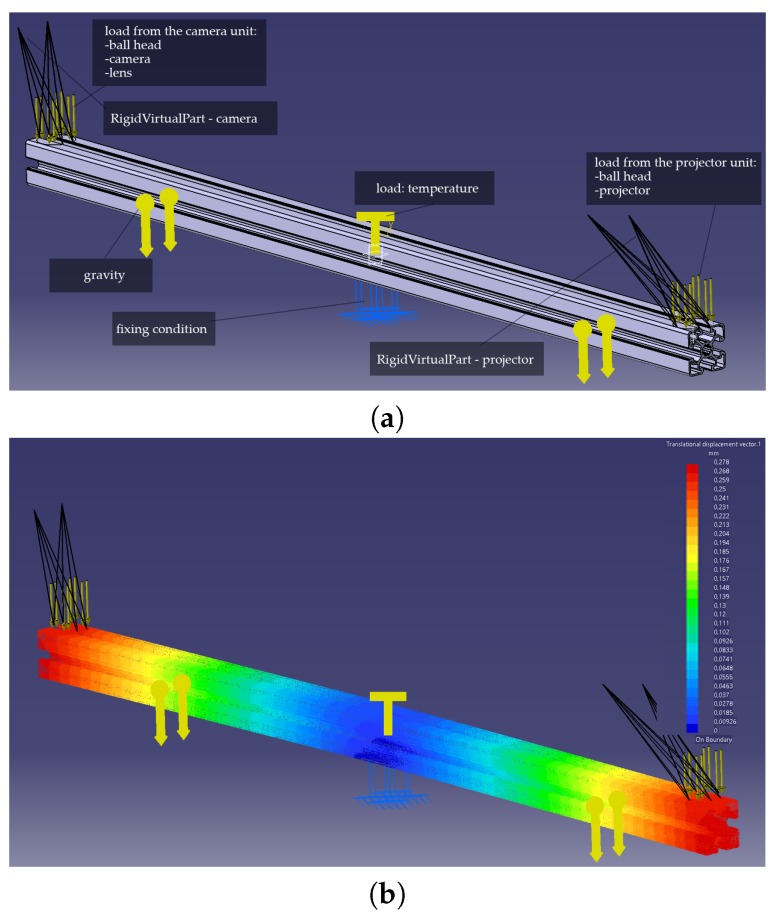
The simulation of the effect of temperature on the mechanical base of a 3D scanner: (**a**) Simulation definition, loads: from the camera and projector unit mass, temperature, gravity, and two RigidVirtualParts; and (**b**) the results from the simulation: thermal deformations of the mechanical base.

**Figure 10 sensors-20-00362-f010:**
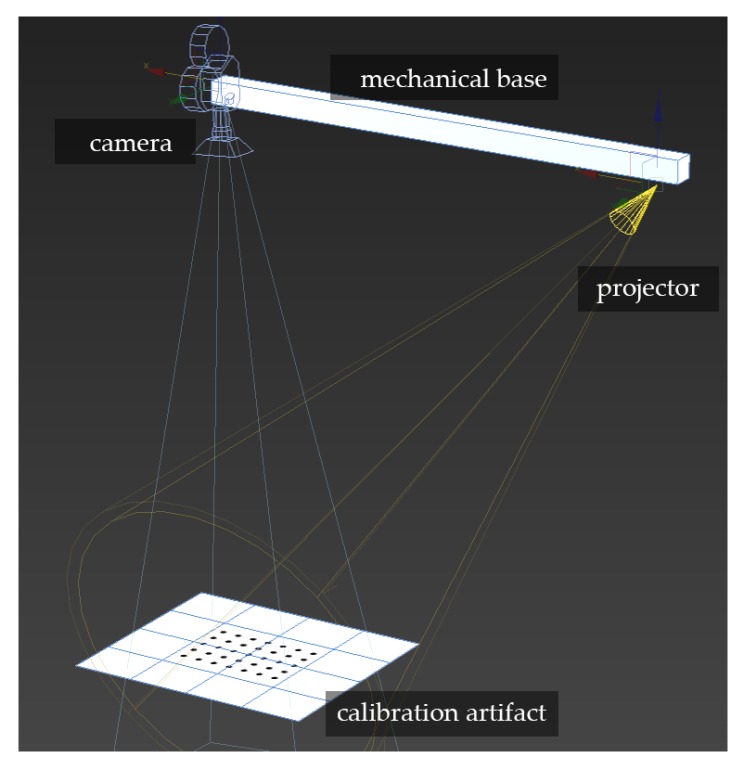
Virtual 3D scanner designed in Autodesk 3DsMAX [62], consisting of a physical camera and light source mapped with a bitmap which projects structured light on the measurement scene.

**Figure 11 sensors-20-00362-f011:**
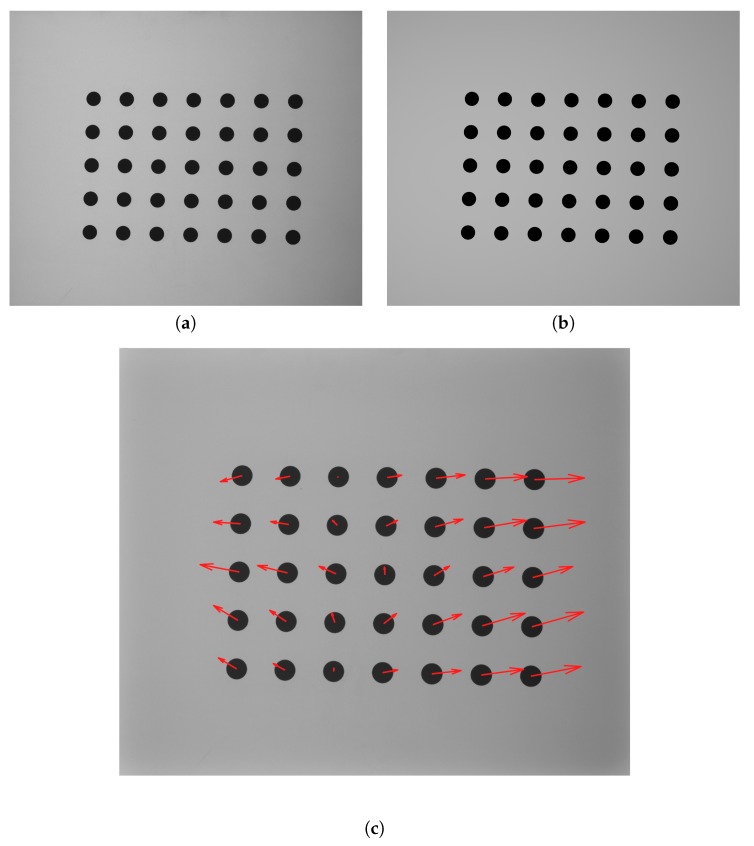
Comparison of the frame captured by the camera and that rendered in the designed virtual scene in 3DsMAX: (**a**) the actual image of the calibration artifact captured by the camera from the 3D scanner; and (**b**) the same image rendered in the 3DsMAX environment using the reconstructed geometry of the 3D scanner; (**c**) a map of marker’s centers shifts (between real and rendered frame) presented by vectors, each vector length was multiplied × 100, and maximum value of shift was no higher than 0.5 px.

**Figure 12 sensors-20-00362-f012:**
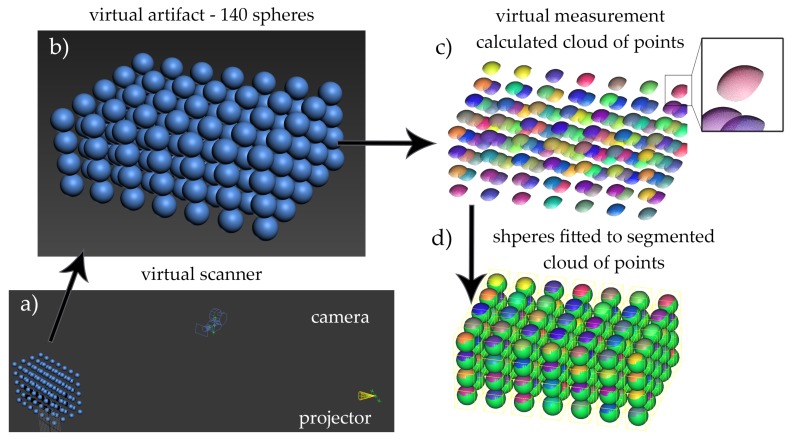
Workflow of the virtual scanner: (**a**) the virtual validation artifact consisting of 140 balls evenly distributed among the scanner’s measurement volume, (**b**) the artifact is measured by the virtual scanner, in which the parameters are exactly the same as the real calibrated one, (**c**) visualization of virtual clouds of points segmentation—different colors, (**d**) ideal spheres are fitted to each ball segment in the validation artifact.

**Figure 13 sensors-20-00362-f013:**
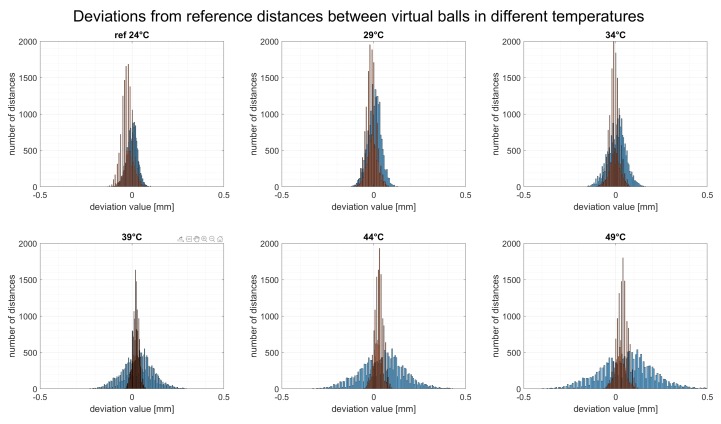
The relative error of all possible distances between each set of 140 virtual balls presented as a set of six histograms. Blue bars shows the distance deviations before compensation and orange after compensation.

**Figure 14 sensors-20-00362-f014:**
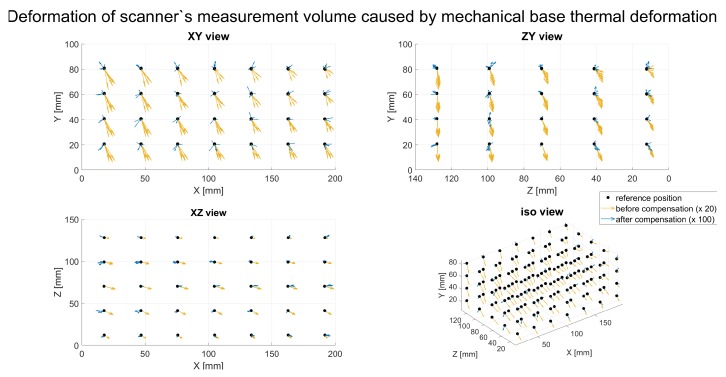
The deformation of scanner measurement volume caused by changing the base temperature, before and after compensation. The yellow vectors represent the direction of displacement of every second virtual ball’s center before compensation caused by a temperature rise of $25{\;}^{\circ }$C; the length of these vectors are multiplied by 20. The blue vectors describe the same directions after compensation (multiplied by 100).

**Figure 15 sensors-20-00362-f015:**
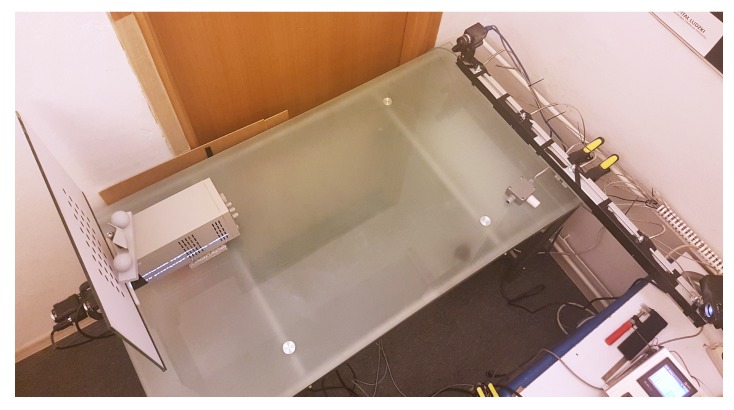
The test stand used in the verification experiments. The 3D-structured light scanner is equipped with a set of heating resistors that can heat its mechanical base. Six temperature sensors measure the temperature of the base. In the measurement volume of the scanner, there is a ball-bar for validation and a calibration artifact in the background.

**Figure 16 sensors-20-00362-f016:**
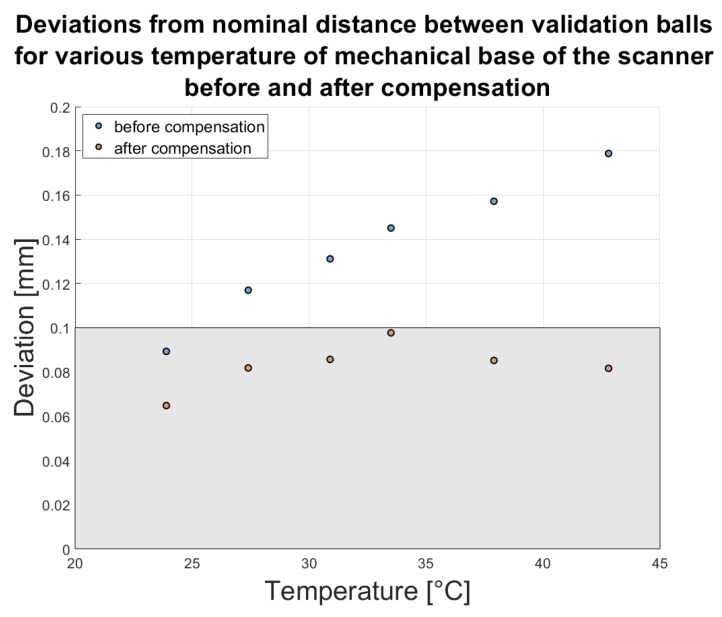
Deviations from the nominal distance between validation balls for various temperatures of the scanner’s mechanical base, before and after compensation. The grey area describes the area where the deviations are no higher than ±0.1 mm.

**Table 1 sensors-20-00362-t001:** Ball-bar artifact dimensions measured on CMM and 3D scanner with reference temperature 24 °C.

CMM measurements: $D_{1}=40.0453\;\mathrm{mm}\;E_{MPE}=\pm 1.8\;\mu m$ $D_{2}=40.0555\;\mathrm{mm}\;E_{MPE}=\pm 1.8\;\mu m$ $d=107.0162\;\mathrm{mm}\;E_{MPE}=\pm 2\;\mu m$ average 3D scanner measurement:						
position	$\bar{D_{1}}$ [mm]	$\bar{D_{2}}$ [mm]	*d* [mm]	$\bar{\Delta D_{1}}$ [mm]	$\bar{\Delta D_{2}}$ [mm]	$\bar{\Delta d}$ [mm]
A	40.0764	40.0342	107.1002	−0.0311	0.0213	−0.0840
B	40.1124	40.1429	107.0245	−0.0671	−0.0874	−0.0083
C	40.1085	40.0741	107.0041	−0.0632	−0.0186	0.01210
D	40.0254	40.0125	107.0925	0.0199	0.0430	−0.0763

**Table 2 sensors-20-00362-t002:** Results of simulation of the effect of temperature on the mechanical base of a 3D scanner. The table contains values of the camera ((**C**) and projector ((**P**) positions ((**pos**) and orientations ((**dir**), at the reference temperature 24 °C, and deviations for five different temperatures. Dimensions are in millimeters.

		Reference $24{\;}^{\circ }$C			$29{\;}^{\circ }$C			$34{\;}^{\circ }$C			$39{\;}^{\circ }$C			$44{\;}^{\circ }$C			$49{\;}^{\circ }$C		
		X	Y	Z	ΔX	ΔY	ΔZ	ΔX	ΔY	ΔZ	ΔX	ΔY	ΔZ	ΔX	ΔY	ΔZ	ΔX	ΔY	ΔZ
C	pos [mm]	194.1340	11.8470	1127.5750	−0.0531	0.0177	0.0009	−0.1063	0.0355	0.0018	−0.1594	0.0533	0.0027	−0.2126	0.0711	0.0036	−0.2657	0.0888	0.0045
	dir [rad]	−6.7730	4.7650	−79.5700	0.0001	0.0004	0.0000	0.0001	0.0011	0.0001	0.0002	0.0018	0.0001	0.0002	0.0025	0.0001	0.0003	0.0032	0.0002
P	pos [mm]	−735.0240	4.2100	1072.2960	0.0530	0.0178	0.0007	0.1060	0.0356	0.0015	0.1590	0.0535	0.0022	0.2120	0.0713	0.0029	0.2650	0.0892	0.0036
	dir [rad]	50.7280	4.0150	−61.7290	−0.0001	0.0017	0.0000	−0.0002	0.0035	0.0001	−0.0002	0.0051	0.0001	−0.0003	0.0069	0.0002	−0.0004	0.0085	0.0002

**Table 3 sensors-20-00362-t003:** Distances between validation balls of a ball-bar artifact for different temperatures of scanner’s mechanical base, before and after applying compensation model. The deviation is related to the reference measurement $d_{ref}=107.0162$ mm ±1.8μm performed with CMM; see also Section 3 and Figure 8.

Temp [°C]	Before Compensation		After Compensation	
	Distance [mm]	Deviation [mm]	Distance [mm]	Deviation [mm]
23.9	106.9269	0.0892	106.9514	0.0648
27.4	106.8992	0.1169	106.9344	0.0818
30.9	106.8851	0.1311	106.9306	0.0856
33.5	106.8711	0.1450	106.9185	0.0977
37.9	106.8590	0.1571	106.9310	0.0852
42.8	106.8374	0.1788	106.9346	0.0816

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
