# Peer review of "Temperature Compensation Method for Mechanical Base of 3D-Structured Light Scanners"

_sensors, 2020, doi:10.3390/s20020362_

Round 1
Reviewer 1 Report
The authors presented a temperature compensation method to effectively reduce the influence of temperature on the mechanical base of a 3D scanner. The advantage is that the proposed temperature software compensation method is suitable for any existing scanners without additional equipment. Some experiments have been carried out to test the proposed method. However, the paper requires significant revision and additional work, as listed in the following.
The units of position and direction in Table 2 are missed. Similarly, the coordinates units in Figure 15 are missed as well. Please make sure each unit is described/defined. The authors only mentioned the calibration of the real 3D structured light scanning, but did not list the calibration results. Please provide the calibration results in the experimental section. The author mentioned that the reference temperature is 24 degrees. However, this paper only considered the effect of thermal expansion brought by temperature increasing, but does not consider the phenomenon of cold shrinkage brought by temperature drop. Whether the temperature can be reduced to below 24 degrees and to get the corresponding simulation results. This paper analyzes the influence of temperature on the measurement results, but the mentioned compensation model in Section 6 is too short. The paper should focus on the compensation method. Please provide a clearer explanation. Some parts are not clear. For example, why the maximum value of horizontal ordinate in Figure 13 and Figure 14 is 8000mm. Please provide some explanations. At the end of the paper, several features that may limit the application of the compensation model are listed. These details should be considered in the proposed model as much as possible because the proposed temperature compensation model is too ideal.Author Response
The response is loaded in separate docx file.

Reviewer 2 Report
This paper considered an important problem with the temperature effect for 3D structured light scanners. The manuscript is organized well. The proposed method is also useful for the existing scanners. I have several questions.
How much measurement accuracy is lost due to the temperature effects? The measurement errors maybe come from multiple factors, such as system assembly. How do you know which part is from the temperature effects? If possible, please quantize them. What factors produce high temperatures? High-speed motion or working conditions? In experiments, Table 3 shows the before and after compensation results. How do you which one is correct because there isn’t a reference value? In experiments, please provide comparison results using other methods. How about the compensation accuracy? If possible, please explain. What materials are used for the Mechanical base? Do you consider the material on how to provide the temperature? If there is a temperature controller for the mechanical base, the compensation method is still needed or not? In Fig.11, two figures are the same. Please show the differences. Fig.12 is a difficult understanding. In the sub-figure (bottom left corner), what relationship of several components? In other sub-figures, what meaning of different colors?Author Response
The response is loaded in separate docx file.

Round 2
Reviewer 1 Report
I read the original manuscript several weeks ago. The authors have improved the contents and made the corresponding changes according to my suggestions. Therefore, I don’t have further comments on the revised version and agree to accept this paper by Sensors.